# Impact of Air Exposure Time on the Water Contact Angles of Daily Disposable Silicone Hydrogels

**DOI:** 10.3390/ijms20061313

**Published:** 2019-03-15

**Authors:** Petar Eftimov, Norihiko Yokoi, Nikola Peev, Georgi As. Georgiev

**Affiliations:** 1Department of Cytology, Histology and Embryology, Faculty of Biology, St. Kliment Ohridski University of Sofia, Sofia 1164, Bulgaria; peftimov@uni-sofia.bg; 2Department of Ophthalmology, Kyoto Prefectural University of Medicine, Kyoto 602-8566, Japan; nyokoi@koto.kpu-m.ac.jp; 3Department of Optics and Spectroscopy, Faculty of Physics, St. Kliment Ohridski University of Sofia, Sofia 1164, Bulgaria; peev_nikola@abv.bg

**Keywords:** silicone hydrogels, wettability, tear film stability, desiccation, daily disposable contact lens, water gradient

## Abstract

The wettability of silicone hydrogel (SiHy) contact lens (CLs) is crucial for the pre-lens tear film stability throughout the day. Therefore, sessile drop and captive bubble setups were used to study the advancing and receding water contact angles (CA) of four SiHy materials: narafilcon A (TE), senofilcon A (AOD), stenfilcon A (MD), and delefilcon A (DT). TE and AOD have 48% and 38% water content, respectively, and no surface coating. MD (54% water) implements “smart chemistry” with just 4.4% bulk silicone content, while DT has >80% water at its surface. These SiHy were subjected to continuous blink-like air exposure (10 s)/rehydration (1s) cycles for 0, 1, 2, 3, 4, 6, 8, 10, 12, 14, and 16 h. The advancing CA, which measures the rehydration propensity of the CL surface, proved to be the most sensitive parameter to discriminate between the samples. The order of performance for the entire time scale was DT > MD >> AOD ≥ TE. The extended desiccation/rehydration cycling increased the differences between the CA of DT and MD compared to AOD and TE. This suggests that the low Si surface content and the high surface hydration are major determinants of SiHy wettability.

## 1. Introduction

Pre-lens tear film (PLTF) is essential for the comfort of the wearers of daily disposable silicone hydrogel (SiHy) contact lens (CLs), as it ensures the lubricity and the optical quality (i.e., the visual clarity and the refractive index) of the CL throughout the day [1]. In turn, the CLs properties, in particular their wettability, play a vital role in the stability of PLTF. It is well known that if the hydration of the CL surface becomes compromised, then PLTF gets unstable due to dewetting [2,3].

The challenges to the maintenance of long term CL wettability at the ocular surface are inherent to the very structure of the silicone hydrogel contact lens [4]. SiHy CLs represent a composite of polymer materials with: (i) a surface covered by hydrophilic chains aimed to render it wettable and (ii) a hydrophobic silicone rich core ensuring the oxygen transmissibility of the material. However, in the course of extended (few hours) wear, the CL is exposed to continuous cycles of air drying (open eye) and rapid rehydration (eye closing at blink). The accumulating desiccation stress modifies the sample structure and makes it possible for the silicone moieties to migrate from the core to the surface of the CL and to impair its properties [5,6].

Great efforts have been devoted to the design of daily disposable SiHy CLs that can maintain high wettability even after extended wear. Different approaches have been implemented [7,8]: (i) from incorporation of wetting agents (polyvinylpyrrolidone, poloxamers, etc.) in the CL core and in CL solutions to (ii) the design of sophisticated CL surfaces with high water holding properties. A recent example of the latter strategy is the implementation of water gradient technology in Dailies Total 1 with water content that rises from 33% at the core to more than 80% towards the interface [9].

Although wetting agents may play a role in vivo, considering the rapid turnover rate (10.3 ± 3.7%/min) of aqueous tears, their effect may be short term or even if resistant to washout they may not match the performance of true surface coatings or treatments [10,11]. Therefore, it is important to evaluate the wettability of the SiHy CL materials alone (without external desorbable wetting agents) [12]. If SiHy material itself has high wettability, it may contribute to the stability of the PLTF at the challenging and dynamic physiological conditions at the ocular surface even after all exogenous agents are washed out by the aqueous tear turnover [13,14]. Furthermore it is very important to probe what happens with the wettability not only of unstressed hydrated samples (as it is most commonly done), but after the materials are subjected to continuous air exposure/rehydration cycling similar to the one that occurs at the ocular surface during daily wear [15].

Four different silicone hydrogels are selected in the current study: narafilcon A (TE), senofilcon A (AOD), stenfilcon A (MD), and delefilcon A (DT). TE and AOD are SiHy materials with 48% and 38% water content, respectively, and no surface coating; polyvinylpyrrolidone based internal wetting agent is utilized instead. MD (54% water content) implements the so called “smart chemistry” with just 4.4% silicone content, while DT utilizes water gradient technology resulting in >80% water content at the CL surface [13,14]. These SiHy were subjected to continuous cycles of air exposure (10 s) and rapid rehydration (1s) similar to the blink dynamics at the ocular surface for 0 (no air exposure at all), 1, 2, 3, 4, 6, 8, 10, 12, 14, and 16 h.

The wettability of the samples is evaluated by the measurement of the advancing and receding water contact angles (CA) via (i) sessile drop and (ii) captive bubble (free and confined) techniques, both in static and dynamic (i.e., contraction and expansion of the bubble) modes. Advancing contact angle reports on the affinity of water to hydrate the surface of SiHy after pre-exposure to air (to desiccation), while the receding contact angle accesses the interaction (the “water holding” property of the CL) of the retreating water front with the hydrated CL surface [6,12]. Water contact angles determined by various techniques have been shown to provide reliable in vitro estimate to the wettability of CLs with broad range of water content and polymer compositions [2,4,6,12,13,14,15,16,17]. The capability of the different approaches to the measurement of CA (with drops or bubbles; in static or dynamic mode) to differentiate the studied SiHy materials is also analyzed.

## 2. Results

The dependencies of the advancing water contact angle of sessile drops over the SiHy materials on the duration of the blink like desiccation/rehydration cycling are presented at Figure 1 (ANOVA and post-hoc pairwise comparisons between the samples at each time point are summarized in Appendix A).

It can be seen that the materials showed very different performance. TE and AOD displayed CA of 66.7° and 68.6°, even prior to air exposure, and the CA rapidly grew to 83.3° (AOD) and 87° (TE). Then for both CLs the contact angle increased slowly to reach 94.5° for AOD and 93.3° for TE after 16 h of desiccation/rehydration cycling.

The advancing water CAs of DT and MD were significantly lower than the ones of TE and AOD for the entire time scale of exposure to desiccation/rehydration cycling. It can be seen that for each time point DT showed significantly lower contact angles than MD with the difference between the samples increasing with the accumulation of exposure to desiccation. It should be noted that in contrast to TE and AOD which showed similar CA/time curves, the shape of the advancing contact angle transients was very different between DT and MD. For MD the CA of the fresh sample was 32.1° which rose to 47.2° at 3 h of cycling; then the CA was relatively stable up to 6 h after which it started to increase steeply to reach 89° after 16 h of cycling. The fresh DT samples displayed 23.5° CA which rose to 38° at 3 h and then grew only gradually to 49.3° after 16 h.

In order to statistically compare the differences in the advancing CA observed between the different pairs of CLs specimen at the various time points, Cohen’s d (Equation(1)) was utilized [18,19,20] (also post-hoc pairwise comparisons with Tukey–Kramer test are summarized in Appendix A):(1)$$d=\frac{\left|\bar{X_{1}}-\bar{X_{2}}\right|}{\sqrt{MS_{within}}}$$
where: $\left|\bar{X_{1}}-\bar{X_{2}}\right|$ is the absolute value of the differences between the group means; the root square of MS_within_ = (n − 1)(SD_1_^2^) + (n − 1)(SD_2_^2^) + … provides measure of the pooled standard deviation (SD) of the multiple samples analyzed (*n* = sample size). The standard deviation (σ(d)) of Cohen’s d is calculated as: σ(d) = [(n_1_ + n_2_)/(n_1_n_2_) + d^2^/2(n_1_ + n_2_)]^1/2^ where n_1_ = n_2_ = 10 is sample size.

Thus Cohen’s d represents a measure of the effect size, i.e., the normalized difference between the mean wettability of two samples, which accounts not only for the differences in the advancing CA, but also for the sample size and the magnitude of the noise (the random error) among the individual measurements within a sample [18,19,20]. It allows for visual and numerical estimation between the differences in the properties of each pair of SiHy materials and also for evaluation of the significance of the observed effect size (d ≥ 0.8—large; d = 0.5—medium; and d = 0.2—small).

As shown at Figure 2, although the effect sizes for almost all SiHy pairs were large, the superiority of DT to the rest of the samples was particularly high. The d(t) curves of DT vs. AOD and DT vs. TE almost overlapped and showed d > 75 at 3 h which gradually decreased to d = 23 at 8 h and then smoothly increased to a plateau of 38 at ≥14 h. The DT vs. MD comparison showed that at ≤8 h Cohen’s d adopted values within the range 5–15, and then it continuously increased to reach d = 33 at 16 h.

The comparison of MD vs. AOD and MD vs. TE revealed very high values of d (up to 75 at 2 h) for the first hours of desiccation/rehydration cycling which at ≥3 h started to decrease and reached d = 3 at 16 h. The AOD vs TE comparisons revealed much lower d values for the entire time scale, with many of the points corresponding to only medium or small effect size magnitude.

As Figure 3A shows it was not possible to obtain quantitative data on the receding water CA of DT and MD with free bubble (i.e., a bubble detached from the syringe needle tip). The reason was that although various modifications of the experimental protocol were tried for most of the time points, the free bubble did not stably adhere to these CL. Still the experiment provides useful qualitative information: if the free bubble stably adhered to the surface of AOD and TE (see movie *CBCA static.avi* in Appendix A), in the case of DT and MD for most of the time points the bubble either resisted to attach to the CL surface (typical for fresh DT; see *CB nonattaching.avi* in Appendix A) or promptly rolled out of the CL surface (see *CB rolling.avi* in Appendix A). Similar impossibility to attach the bubble to well hydrated CL surface was also previously reported and may explain why this approach was used in few studies and experiments with bubble attached to syringe needle are most commonly performed [21,22].

The data on the receding water contact angle of needle confined bubble are summarized in Figure 3B (multiple post-hoc pairwise comparisons at each time point are summarized at Appendix A). The difference in the CA values obtained with free bubble and bubble attached to needle tip are well known and attributed to the different modes of measurement implemented in both protocols [21,22]. As previously reported the differences in the receding CA of SiHy materials were much lower than the ones found with receding CA.

It can be seen that TE showed significantly higher CA compared to the rest of the CLs at all the time points. The other three CL materials performed very similarly to each other for 1–4 h of desiccation/rehydration cycling. Then for the rest of the time points there is very significant difference between the samples, with DT showing the lowest receding CA and MD performing superiorly to AOD (after the 6 h), which in turn approximated the performance of TE. The statistical evaluations (Appendix A) clearly confirmed that at >4 h of cycling the differences between most of the SiHy samples were highly significant. The time dependence of DT receding CA showed a distinct shape: after rising from 19.3° (fresh samples) to 37° at 4 h, the CA remained almost unchanged for eight more hours, and only after that it gradually rose to 48.4° at 16 h.

Figure 4, reveals that although Cohen’s d for most of the SiHy pairwise comparisons were large (>0.8), the effect sizes were much smaller in the case of receding contact angle (d was always <10), compared to d when advancing CAs were analyzed (where values as high as 80 were observed). DT was superior to all other SiHy but to a much lesser extent compared to the advancing CA property. Interesting behavior was observed in case of MD and AOD. In contrast to their performance in advanced CA studies here for the first 4 h both SiHy performed very similarly (d is insignificant at 3 h and 4 h), but at further desiccation/rehydration cycling d started to grow due to the superiority of MD while AOD deteriorated and its performance become very similar to the one of TE.

As demonstrated previously [23,24,25], the advancing water CA (Figure 1) can be used to calculate the adhesion tension, i.e., the propensity of a liquid (the water solution) attraction toward the silicone hydrogel surface. This is done by Equation (2):cos θ = (γ_SV_− γ_SL_)/γ_LV_(2)
where: cos θ—cosine of the advancing water CA; γ_SV_, γ_SL_ and γ_LV_ are the interfacial tensions of the solid/vapor, solid/liquid and liquid/vapor interfaces respectively (γ_LV_ = 72.9 mN/m for the aqueous buffer/air surface at normal temperature). The term (γ_SV_ − γ_SL_) represents the adhesion energy. The higher it is, the higher is the affinity of the aqueous solvent to hydrate the contact lens surface.

As can be seen at Figure 5, for AOD and TE the adhesion energy was <30 mN/m even for fresh (unexposed to desiccation) samples. It steeply decreased at 3 h to 8.47 mN/m and 3.72 mN/m for AOD and TE, respectively, and then gradually diminished to reach negative values for both materials at ≥12 h desiccation/rehydration cycling. The negative hydration energy reflects that the advancing water CA for both materials had reached values >90° which is characteristic for samples with prevalence of hydrophobic, water repellent patches at their surface. MD showed adhesion energy as high as 61.75 mN/m at 0h which first gradually diminished to 49.55 mN/m at 3 h and at ≥6 h started to steeply decrease with further desiccation/rehydration cycling to reach 1.51 mN/m at 16 h. The temporal pattern of DT adhesion energy was very distinct from the rest of the samples. DT had adhesion energy of 66.87 mN/m prior exposure to desiccation (0 h) which proved much more resistant to desiccation/rehydration cycling and remained as high as 47.48 mN/m after 16 h of desiccation/rehydration treatment.

## 3. Discussion

The results demonstrate the better wettability (i.e., the lower water contact angles) of DT and MD compared to TE and AOD for the entire time scale of exposure to desiccation stress. Such outcome is in agreement with the data on the composition and structure of the SiHy samples. MD implements “smart chemistry” allowing for merely 4.4% silicon content in the specimen. In contrast for the rest of the materials studied, the bulk content of the Si-rich hydrophobic phase is estimated to be ≥30% [26]. In the case of DT, the influence of the hydrophobic phase on the CL wettability is neutralized by the utilization of water gradient technology resulting in >80% water at the outer CL surface [9]. The statistically significant superiority of DT over MD for most time points of desiccation stress exposure emphasizes that the high interfacial content of water (and the low one of Si) may be more critical for the CL wettability than the bulk amount of silicone. This is also illustrated with the much stronger increase of the advancing CA of MD compared to DT at ≥6 h exposure to desiccation/rehydration cycling. The high water contact angles of TE and AOD can be explained with: (i) limited ability of the internal wetting agents (PVP based) to protect the CL surface as compared to surface coatings and (ii) the lower water content and the higher silicone content of these materials that are supposed to result in relatively high surface concentration of silicone as well. The latter was well illustrated by a recent study using X-ray photoelectron spectroscopy to probe the Si content in the outermost 3 nm of CLs exposed to dryness [27]. It was found that the Si surface concentration of narafilcon A and senofilcon A was 8.5% and 12.8%, respectively. In contrast, the outermost region of delefilcon A contained merely 0.5% Si. It is interesting that in spite of the “smart chemistry” technology and the very low bulk percentage of silicon, the surface content of Si of the desiccated MD sample was as high as 10.2%. Thus the importance of true coating for the long term control of CL surface properties is emphasized as compared to implementation of (internal) wetting agents and to alterations of bulk Si content. These findings align very well with the high advancing CA of MD after prolonged air exposure/water immersion cycling with values >80° at the 16 h (vs 48° for DT).

It should be noticed that both in our control experiments (data not shown) and in publications by independent teams TE and AOD type of materials were found to display advancing contact angles ≥47° even in absence of desiccation stress and with all the wetting agents being present (i.e., with the CLs freshly removed from blisters and without exposure to air) [7,21,22,28,29]. Considering that it has been claimed [30] that the internal wetting agents infused in the core of senofilcon A and narafilcon A are not blink released (i.e., significantly resisting a depletion by the aqueous tear turnover) it is further indication on the importance of true surface coatings/treatments for the low water CA of CL. Still, in order to evaluate whether there is significant correlation between the water contact angles of pure (with exogenous wetting agents removed) silicone hydrogels and the in vivo “in eye” performance of the materials more data are necessary on the CA of worn (at the end of the day) CLs with highly wettable surfaces (like DT and MD) and the corresponding clinical estimates (PLTF stability, patient comfort indices, etc.) of the CL performance [1,14,28]. The data on the impact of internal wetting agents on the contact angles of worn SiHy CLs are also inconclusive [7,8] and although it was found that the presence of PVP (wetting agent) does not significantly improve the clinical performance of SiHy [31], the small sample size (40 patients per CL sample) limited the statistical power of the study.

Apart from the chemistry of the SiHy, the structure of the CL surface might also impact its CA. Indeed, although immediately after removal from the blister solutions, CL materials are found to have a relatively smooth surface, the accumulation of the dehydration stress in the course of dehydration/rehydration cycling can also modify the roughness of the CLs, which has strong impact on wettability by itself [32,33]. The effect can be additionally enhanced in vivo where the deposits of tear film constituents (lipids, proteins, mucins, etc.) are well known to alter both the composition and the structure of the CL surface [34], and correlates with our findings on very different TF breakup patterns of SiHy with internal wetting agents and no surface coating and on contact lens with true surface coating (representative patterns are provided at Appendix A). This suggests that various effects, including both surface chemistry and alterations in surface roughness, may simultaneously contribute to the wettability of the CL surface.

Various techniques are proposed to measure the advancing and receding water contact angles and were also adopted here: (i) from static measurements (sessile drop and free captive bubble) to (ii) dynamic measurements involving compression and expansion of the needle confined bubble [2,4,6,12,13,14,15,16,17]. However, there are almost no comparisons made on the capability of the different approaches to efficiently discriminate materials based on their distinct wettability. The advancing contact angle measures the affinity of an advancing water front to hydrate a surface pre-exposed to dryness (to air), i.e., it accesses the rehydration propensity of a material [6,12]. The receding contact angle (measured when the water is forced to retract from a surface) evaluates the capability of a hydrated material to withhold water at exposure to dryness [6,12]. It was found that the advancing CA can be reliably evaluated via the sessile drop technique while in order to quantitatively estimate the receding CA dynamic measurements with needle confined captive bubble are needed (the free bubble test provided qualitative albeit useful information).

Another important outcome is that the advancing contact angle provides stronger discrimination of the different silicone hydrogel materials compared to receding contact angle over the entire scale of blink like cycling. This aligns with the different states of the SiHy surface in both types of experiments [2,3,28,31]. In receding CA measurements the water front retreats over hydrated sample and in such condition most modern CLs materials perform well and similarly or sometimes identically to each other as reported in multiple studies [12,16,21,22]. As can be seen, for most of the time points, the receding CA of the specimen was lower than the advancing CA. In contrast in the advancing CA experiments, the water front progresses over the desiccation exposed surface and it is precisely this type of condition in which the difference in the material properties of the silicone hydrogel materials becomes the major determinant of wettability. It is thought that a similar state of the CL surfaces occurs at the eye due to the accumulating effect of the exposure to air after few hours of CL wear. Therefore the advancing CA allows for a simple way to probe the interaction of water (or more complex aqueous solutions) with CL materials in vitro. Furthermore, it enables one to evaluate (Figure 5) the adhesion energy, i.e., a direct measure of the propensity of a liquid attraction toward the silicone hydrogel surface [23,24]. Thus, the sessile drop technique provides a reliable and robust methodology to access CL wettability, which can be very important for rapid in vitro high yield screening of multiple SiHy at the early stages of preclinical development of silicone hydrogels. It can be seen that both the advancing contact angle and the adhesion energy were highly sensitive to the SiHy material properties and to the duration of the desiccation/rehydration cycling with decrease of wettability (manifested as raise in advancing CA and decrease in the adhesion energy) with the extension of air exposure. The latter result emphasizes how important is to account for the impact of dryness on CL wettability [15]. In many studies, only freshly hydrated (without air exposure) samples are used that may not account for the different susceptibility of SiHy materials to desiccation stress accumulating in the course of the CL wear at the ocular surface.

The current study represents a pilot evaluation of the time dependent wettability of range of SiHy materials being exposed to blink like desiccation cycling. Although the inherent wettability of a silicone hydrogel bears promise for its performance in vivo, it should be kept in mind that apart from desiccation, at the ocular surface CLs are exposed to the myriad of lipid and protein compounds of the natural tears. These constituents may form deposits at the lens surface and further complicate the wettability pattern of the SiHy materials [2,14,28,35]. Other factor that may also play role in vivo is that the posterior side of the CL is in permanent contact with the post-lens tear film [36]. Its capability to alter the hydration of the CL anterior side by diffusion of water through the hydrophobic silicon-rich lens core or by other mechanisms of tear exchange is hard to estimate as it greatly depends on the CL material properties and on the environmental conditions (air humidity, temperature, wind speed, etc.) and is thought to be limited, especially few hours after the fitting of the CL [36,37]. It is found that the volume of the post-lens fluid squeezed out during the blink or lost due to pervaporation through the CL exceeds the amount of fluid drawn back under the lens at blink [36,38]. This coincides with experimental findings, where the postlens TF thickness is greatly reduced in the course of SiHy wear [39,40]. Therefore in order to see whether the dependence of advancing CA of water (or of a more complex tear mimicking solution) on the air exposure time significantly correlates with the clinical performance of the CL, the in vitro results should be collated with the CA values of the worn CLs and with the corresponding data on the PLTF stability and the patient comfort. Such studies can identify key material properties that are readily accessible in vitro and can serve as an early stage predictor for the physiological “in eye” performance of the silicone hydrogels.

## 4. Materials and Methods

### 4.1. Materials

The SiHy samples used were obtained in the form of commercial contact lens summarized in Table 1. 

Phosphate Buffer Saline (PBS pH 7.4) was used for all measurements as done in previous studies as it mimics the osmolarity and the pH of the aqueous tears [12,22]. Prior contact angle measurements the CL was removed from blister and soaked/washed in PBS (pH 7.4) for 24 h in order to remove wetting agents from blister solution or agents that can be released from the lens core. The absence of desorbable wetting agents was accessed by monitoring of the surface tension of the washing solution [12,21,22,24]; control experiments when the washing period was extended for 48 h did not change the CA values.

Then the CLs were positioned on a dipping machine and subjected to continuous automatic desiccation/rehydration cycling for 0 (no air exposure at all), 1, 2, 3, 4, 6, 8, 10, 12, 14, and 16 h as previously described [15]. Each cycle consisted of 10 s air exposure of the CL and 1 s immersion in PBS buffer. This regime was chosen to emulate the natural blinking dynamics where interblink time (i.e., the period of time for which the eye is kept open) is reported to vary between 6 s (relaxed condition) to 40 s (when working on mobile phone/computer screen) [15]. For each time point 10 samples of each SiHy material were tested with each of the contact angle measurement protocols (see next point). When the samples were used in sessile drop measurements the CL was taken from the washing solution and repeatedly placed with the test (front) surface in contact with a Supraclean microfibre cloth (Pentax UK, Slough, UK) until any excess surface liquid had been removed [22].

### 4.2. Contact Angle Measurements

The sessile drop and captive bubble contact angle measurements were performed with Contact Angle Meter with Rotatable Substrate Holder, Automated Dispenser & Temperature Control HO-IAD-CAM-01B (Holmarc Opto-mechatronics, Kochi, India). Contact lens was fitted over curved ceramic substrate (matching the CL curvature) and positioned in glass chamber maintaining quiescent environment. The sessile drop and the captive bubble were generated and manipulated by the automatic syringe. The contact angles were measured with HO-IAD-CAM-01B software utilizing the β-spline method [41] which allows to measure the contact angle between the bubble and the curved surface of the contact lens (Figure 6).

Three types of contact angle experiments were performed.

#### 4.2.1. Measurement of Advancing Water Contact Angle with Sessile Drop Method

The lens holder was positioned directly beneath the dosing needle of the microsyringe on the sample holder stage of the HO-IAD-CAM-01B. A 3 µL drop of PBS was formed on the tip of the dosing needle. The stage was elevated until the water drop and the lens surface made contact, at which point the stage was lowered away from the needle. Immediately, a 10 s digital movie clip of the water drop on the lens surface was recorded at a rate of 10 frames per second and at a resolution of 1280 × 960 pixels. The data for the last three seconds were used to estimate the CA as reported previously [22].

#### 4.2.2. Measurement of Receding Water Contact Angle with Free Captive Bubble

Each lens sample was placed onto a custom-built glass curved mount which was inverted and placed into a PBS-filled glass chamber that housed a curved needle from which a 5 µL air bubble was dispensed whilst the lens and the air from the needle were in direct contact. The needle was then retracted, leaving the air bubble at the apex of the lens. After a minute equilibration time a 10 s movie (10 frames per second, 1280 × 960 resolution) was captured after the needle was retracted and its frames were analyzed to obtain the contact angle between the static bubble and the contact lens (see Appendix A
*CBCA static.avi*).

#### 4.2.3. Measurement of Receding Water Contact Angle with Needle Confined Expanding Captive Bubble

A 1 µL air bubble was dispensed from a curved 1.65mm outer diameter blunt-ended needle positioned 2mm directly below the CL apex. The size of the bubble was slowly increased by 0.1 µL/s using the HO-IAD-CAM-01B automated bubble delivery function until contact was made with the CL surface. Assessment of the receding and advancing contact angles was achieved by first enlarging the air bubble at a rate of 0.1 µL/s until it increased in volume by 3 µL and then shrinking its volume until the bubble detached from the lens surface. The typical shape of the time dependences of bubble contact line and water contact angle are presented at Figure 7. The water contact angles were measured as previously described [12,21,22].

Advancing CA showed identical trends as sessile droplet experiments (see Appendix A) but with higher noise (higher SD) inherent to this experimental protocol and were not used here. The receding CA data are presented in the text as explained in the Results section.

### 4.3. Statistical Analyses

Descriptive statistics, one- and two-factor repeated measures ANOVA and multiple pair-wise comparisons (Tukey-Kramer method) were performed with KyPlot 5 (KyensLab, Tokyo, Japan) and PAST 3.22 statistical packages [42,43].

## 5. Conclusions

The wettability was studied of a diverse range of SiHy materials, utilizing internal wetting agents (Senofilcon A and Narafilcon A), “smart” low Si content chemistry (Stenfilcon A) or water gradient technology resulting in 80% water content at the CL surface (Delefilcon A). Various techniques were implemented to measure the advancing and receding water contact angles of the CL: from static measurements (sessile drop and free captive bubble) to dynamic measurements involving compression and expansion of the needle confined bubble. It was found that the advancing contact angle which measures the capability of aqueous tear to hydrate desiccation exposed CL surface proved to be the most sensitive parameter to discriminate between the hydration affinities of the CL samples. As it can be conveniently measured by the sessile drop technique it offers a reliable and robust methodology for rapid in vitro high-yield screening of multiple SiHy at the early stages of preclinical development of CL materials. The advancing contact angle which measures the capability of aqueous tear to hydrate desiccation exposed CL surface proved to be the most sensitive parameter to discriminate between the hydration affinities of the CL samples. The order of performance superiority for the entire time scale studied was Delefilcon A (DAILIES TOTAL 1^®^, Alcon) >Stenfilcon A (MyDay^®^, Cooper Vision) >>Senofilcon A (ACUVUE^®^OASYS^®^ 1-DAY, Johnson&Johnson) ≥ Narafilcon A (1-DAY ACUVUE^®^TruEye^®^, Johnson&Johnson). The accumulation of desiccation stress increased the difference in the performance of Delefilcon A and Stenfilcon A compared to the Senofilcon A and Narafilcon A which suggests that the low Si surface content and the high hydration of the material interface are major determinants of SiHy wettability.

## Figures and Tables

**Figure 1 ijms-20-01313-f001:**
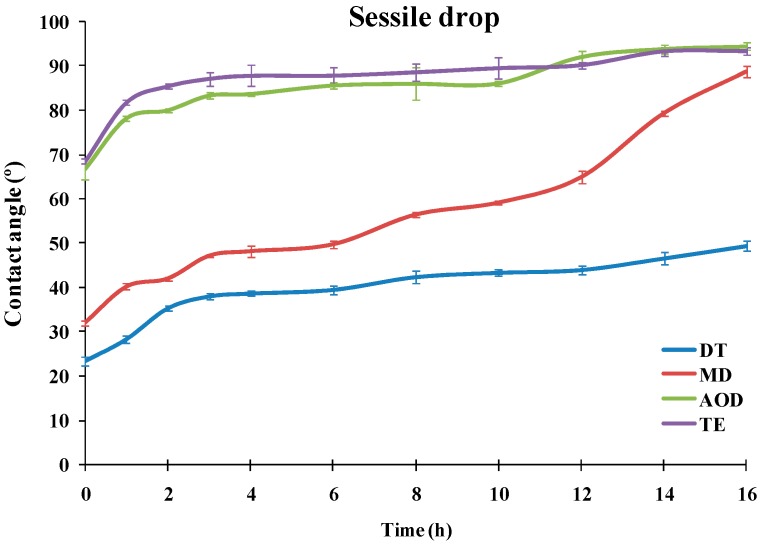
Dependence of the advancing water contact angle of sessile drops over silicone hydrogel materials (*n* =10 for each point) on the duration of the blink like desiccation/rehydration cycling.

**Figure 2 ijms-20-01313-f002:**
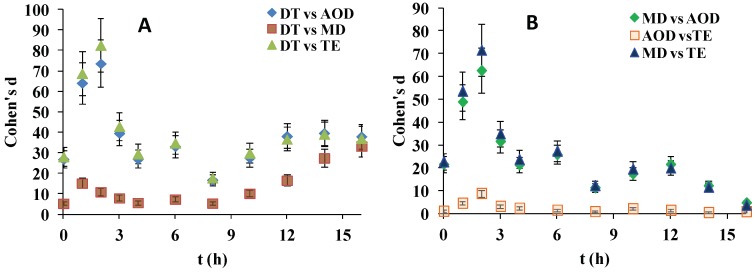
Dependence of Cohen’s d for advancing contact angle on time for pairwise comparisons involving delefilcon A (DT) (**A**) and the rest of the SiHy sample pairs (**B**).

**Figure 3 ijms-20-01313-f003:**
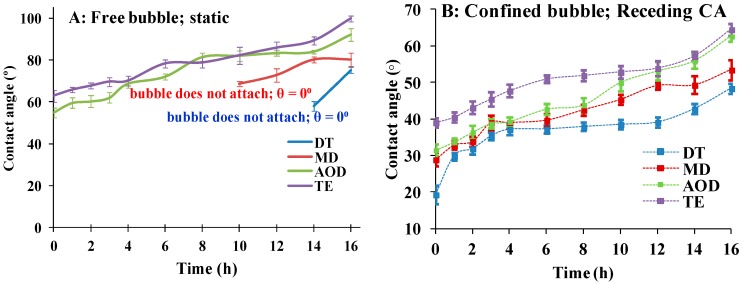
(**A**): Time dependent behavior of receding contact angles of free captive bubbles. As explained in the main text for most of the time points it was not possible to attach the bubble to DT orMD. See the movies *CB nonattaching.avi* and *CB*
*rolling.avi* provided as Appendix A. (**B**): Dependence of receding water contact angle of SiHy materials on the duration of blink like desiccation/rehydration cycling. The experiments were performed with captive bubble confined to the syringe needle tip (*n* = 10 for each point).

**Figure 4 ijms-20-01313-f004:**
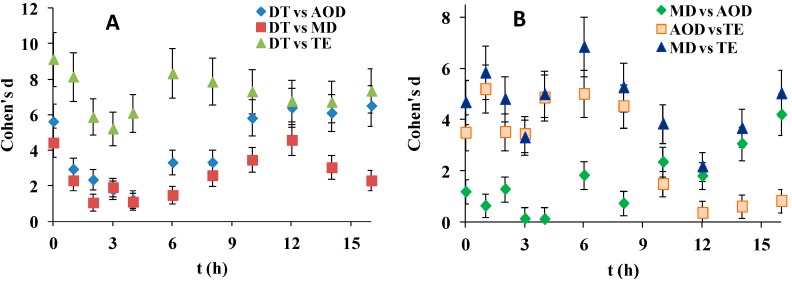
Dependence of Cohen’s d for receding contact angle (data from Figure 3B) on time for pairwise comparisons involving DT (**A**) and the rest of the SiHy pairs (**B**).

**Figure 5 ijms-20-01313-f005:**
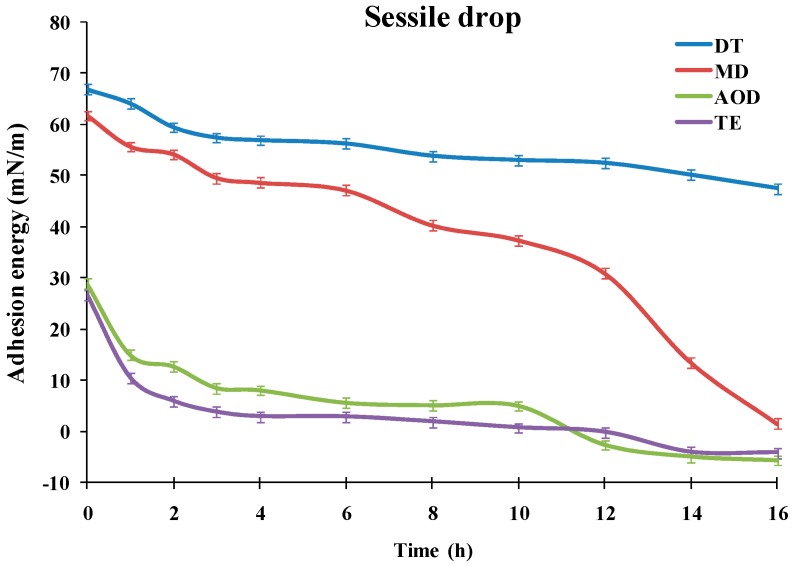
Dependence of the adhesion energy (see Equation (2) in the main text) of SiHy materials (*n* = 10 for each point) on the duration of the blink like desiccation/rehydration cycling.

**Figure 6 ijms-20-01313-f006:**
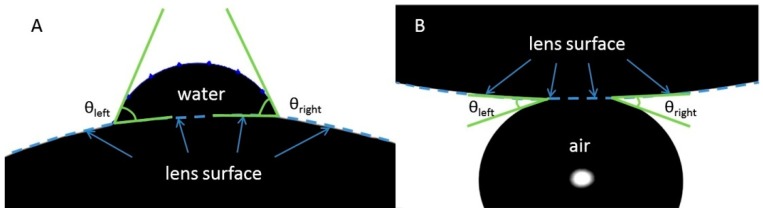
Measurement of advancing (**A**) and receding (**B**) water contact angle in sessile drop and captive bubble configuration respectively.

**Figure 7 ijms-20-01313-f007:**
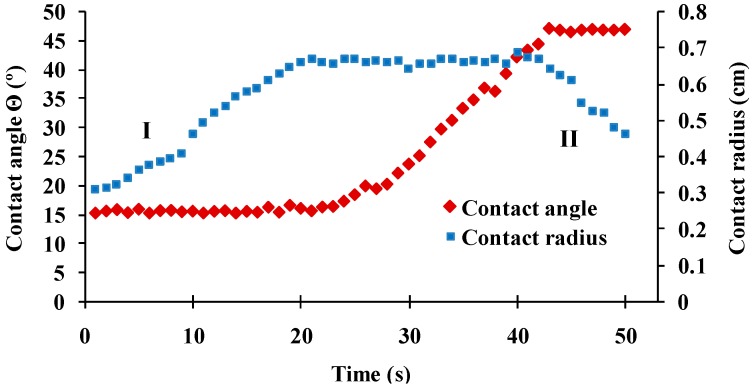
Contact angle and contact diameter vs frame number for a CL sample in captive bubble setup. The receding (**I**) and advancing phases (**II**) are denoted. Frames showing constant contact angle at contact radius expansion are used for receding CA evaluation. Frames showing constant contact angle at contact radius contraction are used for advancing CA evaluation.

**Table 1 ijms-20-01313-t001:** Summary of SiHy materials proprietary names and of some material properties of interest. (Data provided by the manufacturers.).

Specifications	Proprietary Name			
	ACUVUE^®^OASYS^®^ 1-DAY	DAILIES TOTAL 1^®^	MyDay^®^	1-DAY ACUVUE^®^TruEye^®^
**Manufacturer**	*Johnson & Johnson*	*Alcon*	*Cooper Vision*	*Johnson & Johnson*
**USAN**	Senofilcon A	Delefilcon A	Stenfilcon A	Narafilcon A
**Surface treatment**	None. Hydraluxe^®^ internal wetting agent	Water gradient technology	Smart silicone chemistry	None. PVP as internal wetting agent.
**Water content**	38%	33% at core, >80% at surface	54%	48%

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
