# Peer review of "Impact of Air Exposure Time on the Water Contact Angles of Daily Disposable Silicone Hydrogels"

_ijms, 2019, doi:10.3390/ijms20061313_

Reviewer 1 Report

This study measures the wettability as a function of time of four different contact lens' materials belonging to three different companies (J&J, Alcon, and Cooper Vision). 

The major finding is that the material Delefilcon A has the highest wettability. This material is what Alcon lenses are made of and the reader should keep on mind that this study has been funded by Alcon. Nonetheless, the methods used in this article for capturing and analyzing data are robust.

I like the article on its current form and it is a great example of what a good multidisciplinary collaboration can achieve.

My comments to the article are minor:

1- Introduce the abbreviations used in the study in table 3. 

2- The units on the graphs should appear as Variable (unit). ANd not how they appear not Variable, unit.

3- Figures 2 and 3 should become Figures 2a and 2b and be shown side by side.

4- Figure 5 should be replaced. The current figure is obscure and it does not show with clarity the definition of the contact angles for each method. Should we expect to find a direct correlation between an increase on both of the angles and wettability or an inverse relationship, or one direct and one inverse?

5- Figure 6. Moved the series label to a place where are not on top of the x-axis.

6- Find a better way to communicate the information in table 1. The current table is unnecessary.

7- Find a better way to communicate the information in table 2. I would prefer the authors describe in a paragraph that information. The current table position is unnecessary during the body of the paper. If considered strictly necessary by the authors, at least move it to the materials and methods section. 

8- How can be a lens composed of up to almost 100% water? Page 2 line 48. This could be considered outside the lens already.

9- Rewrite the sentence on page 6 lines 156-157.

10- On the conclusion section please label the lenses by their materials, commercial name, and manufacturers. Page 10, lines 342-343. Many people will read the conclusion before the rest of the paper and it will help them to identify the lenses properly.

Congratulations on the work. Looking forward to the reviewed version.

Author Response

Answers to Reviewer 1:

We thank the Reviewer for the high appreciation of our study and for his his valuable time and comments. Enlisted are point by point replies (also provided as separate PDF file):

 1. Introduce the abbreviations used in the study in table 3.

 All abbreviations used are defined in a dedicated table at page 12, line 447.

 2. Formatting changes:

The units on the graphs should appear as Variable (unit). And not how they appear not Variable, unit.

Figures 2 and 3 should become Figures 2a and 2b and be shown side by side.

Figure 6. Moved the series label to a place where are not on top of the x-axis.

 All the formatting changes are now implemented.

 3. On Tables:

- Find a better way to communicate the information in table 1. The current table is unnecessary.

- Find a better way to communicate the information in table 2. I would prefer the authors describe in a paragraph that information. The current table position is unnecessary during the body of the paper. If considered strictly necessary by the authors, at least move it to the materials and methods section.

 Both tables are now moved to Supplement 1. Instead in order to statistically compare the differences in the advancing and receding contact angles observed between the different pairs of CLs specimen at the different time points, Cohen’s d was utilized as recommended for the analysis of laboratory or clinical properties of biomaterials (Ratner, B.D., Correlation, Materials Properties, Statistics and Biomaterials Science, Chapter III.1.3 in Biomaterials Science (Third Edition) An Introduction to Materials in Medicine, Editors: Buddy D. Ratner, Allan S. Hoffman, Frederick J. Schoen, Jack E. Lemons, 2013, p: 1354 and others). Cohen’s d represents a measure of the effect size, i.e. the normalized difference between the mean wettability of two samples, which accounts not only for the differences in the advancing CA, but also for the sample size and the magnitude of the noise (the random error) among the individual measurements within a sample. Thus it allows for visual and numerical estimation between the differences in the properties of each pair of SiHy materials and also for evaluation of the significance of the observed effect size (d≥0.8- large; d= 0.5- medium; d= 0.2- small).

The analysis is presented at lines 99-129 and lines 173-181 and as Fig. 2 and Fig. 4 in the revised manuscript.

 4. Figure 5 should be replaced. The current figure is obscure and it does not show with clarity the definition of the contact angles for each method. Should we expect to find a direct correlation between an increase on both of the angles and wettability or an inverse relationship, or one direct and one inverse?

 The figure is now replaced. The position of the contact lens surface, the water drop/air bubble and of the surrounding media are now precisely denoted so that to allow for unambiguous understanding of the figure.

 4. How can be a lens composed of up to almost 100% water? Page 2 line 48. This could be considered outside the lens already.

 The statement comes from the description provided by the manufacturer but it is definitively speculative and is now removed.

 5. Rewrite the sentence on page 6 lines 156-157.

 The sentence is rewritten.

 6.  On the conclusion section please label the lenses by their materials, commercial name, and manufacturers. Page 10, lines 342-343. Many people will read the conclusion before the rest of the paper and it will help them to identify the lenses properly.

 The Conclusion section now states:

 The wettability was studied of diverse range of SiHy materials, utilizing internal wetting agents (Senofilcon A and Narafilcon A), “smart” low Si content chemistry (Stenfilcon A) or water gradient technology resulting in 80% water content at the CL surface (Delefilcon A). Various techniques were implemented to measure the advancing and receding water contact angles of the CL:  from static measurements (sessile drop and free captive bubble) to dynamic measurements involving compression and expansion of the needle confined bubble. It was found that the advancing contact angle which measures the capability of aqueous tear to hydrate desiccation exposed CL surface proved to be the most sensitive parameter to discriminate between the hydration affinities of the CL samples. As it can be conveniently measured by the sessile drop technique it offers a reliable and robust methodology for rapid in vitro high yield screening of multiple SiHy at the early stages of preclinical development of CL materials. The advancing contact angle which measures the capability of aqueous tear to hydrate desiccation exposed CL surface proved to be the most sensitive parameter to discriminate between the hydration affinities of the CL samples. The order of performance superiority for the entire time scale studied was Delefilcon A (DAILIES TOTAL 1®, Alcon) > Stenfilcon A (MyDay®, Cooper Vision) >> Senofilcon A (ACUVUE®OASYS® 1-DAY, Johnson&Johnson) ≥ Narafilcon A (1-DAY ACUVUE® TruEye®, Johnson&Johnson). The accumulation of desiccation stress increased the difference in the performance of Delefilcon A and Stenfilcon A compared to the Senofilcon A and Narafilcon A which suggests that the low Si surface content and the high hydration of the material interface are major determinants of SiHy wettability.

Reviewer 2 Report

The article:

 „Impact of air exposure time on the water contact 2 angles of daily disposable silicone hydrogels by:

 Petar Eftimov, Norihiko Yokoi, Nikola Peev, and Georgi As. Georgiev,

deals with interesting approach the studies of the wettability of silicone hydrogel (SiHy) contact lens due to for the pre-lens tear film stability throughout the day. The sessile drop and captive bubble setups were used to study the advancing and receding water contact angles of four SiHy materials. As SiH materials for measurement were used following contact lenses: narafilcon A, senofilcon A, stenfilcon A and delefilcon A.

The legitimacy of this research is important because it pre-lens tear film is essential for the comfort of the wearers of daily disposable silicone hydrogel contact lens , as it ensures the lubricity and the optical quality (i.e. the visual  clarity and the refractive index) of the  throughout the day.  The maintenance of long term contact lenses wettability at the ocular surface are inherent to the very structure of the silicone hydrogel contact lenses.

In conclusion the authors state that the advancing contact angle which measures the capability of aqueous tear to hydrate desiccation exposed surface proved to be the most sensitive parameter to discriminate between  the hydration affinities of the contact lens samples. Further the authors suggests that the low Si surface content and the high hydration of the material interface are major determinants of SiHy wettability.

The experimental processes and measurement methods are good presented. The experiments conducted by the authors, calculations, results and discussion are very interesting and correct. Tables and graphical presentation of figures are also correct. So this article surely can be published in International Journal of Molecular Sciences.

But, some minor revisions could be suggested to authors to improve soundness of the paper.

And so I have important critical attention:

 1. In my opinion, the final conclusions are too modest in relation to the data presented in the chapters Results and Discussion. I believe that the authors should more emphasize their results and achievements. The Conclussion chapter should be expanded.

Author Response

Answers to Reviewer 2:

 We thank the Reviewer for the positive appreciation of our study and for his valuable time and comments. Enlisted is our reply (also provided as separate PDF file):

 1. In my opinion, the final conclusions are too modest in relation to the data presented in the chapters Results and Discussion. I believe that the authors should more emphasize their results and achievements. The Conclussion chapter should be expanded.

 The Conclusion is now revised as follows at lines 411-429:

 The wettability was studied of diverse range of SiHy materials, utilizing internal wetting agents (Senofilcon A and Narafilcon A), “smart” low Si content chemistry (Stenfilcon A) or water gradient technology resulting in 80% water content at the CL surface (Delefilcon A). Various techniques were implemented to measure the advancing and receding water contact angles of the CL:  from static measurements (sessile drop and free captive bubble) to dynamic measurements involving compression and expansion of the needle confined bubble. It was found that the advancing contact angle which measures the capability of aqueous tear to hydrate desiccation exposed CL surface proved to be the most sensitive parameter to discriminate between the hydration affinities of the CL samples. As it can be conveniently measured by the sessile drop technique it offers a reliable and robust methodology for rapid in vitro high yield screening of multiple SiHy at the early stages of preclinical development of CL materials. The advancing contact angle which measures the capability of aqueous tear to hydrate desiccation exposed CL surface proved to be the most sensitive parameter to discriminate between the hydration affinities of the CL samples. The order of performance superiority for the entire time scale studied was Delefilcon A (DAILIES TOTAL 1®, Alcon) > Stenfilcon A (MyDay®, Cooper Vision) >> Senofilcon A (ACUVUE®OASYS® 1-DAY, Johnson&Johnson) ≥ Narafilcon A (1-DAY ACUVUE® TruEye®, Johnson&Johnson). The accumulation of desiccation stress increased the difference in the performance of Delefilcon A and Stenfilcon A compared to the Senofilcon A and Narafilcon A which suggests that the low Si surface content and the high hydration of the material interface are major determinants of SiHy wettability.

Reviewer 3 Report

Dear authors, 

the work "Impact of air exposure time on the water contact 2 angles of daily disposable silicone hydrogels" is well written and approach an interesting topic. I found the approach of the wet and dry too simplistic, in fact despite the blinking can be seen as an average of 10 seconds dry, the authors never mention the fact that one side of the CL is always in contact with the tear film. at the end of the discussion you present some of the limitations ( lipids protein, etc) but you don't mention this specific aspect. 
I suggest to specify why you are using different contact angles measurements mode. Which information you get from each one of them? This part should be improved and references of the techiniques should be given. 
The difference in the wettability results are solely due to the chemistry of the material? what about the morphology, to make the work more complete, I suggest a morphological study of the surface ( SEM or AFM) to complete the results

Line 257 , specify the reference ("previous studies")

line 258 did you verify that the 24 hours were enough for the "washing"? 

I suggest to move table 1 and 2 to anexe
figure 6 is misplaced. 

I suggest to add more references of previous studies dealing with silicon CL wettability in the introduction and in the results discussion. There are several of them (example " Controlled Release of Antibiotics From Vitamin E Loaded Silicone-Hydrogel Contact Lenses " they measure the wettability of AOD and TE, and their results are close to yours for example). I suggest the insertion of More references on what concerns the adhesion energy.

Author Response

Answers to Reviewer 3:

 We thank the Reviewer for his valuable time and comments. Enlisted are point by point replies (also provided as separate PDF file):

 1. The work "Impact of air exposure time on the water contact 2 angles of daily disposable silicone hydrogels" is well written and approach an interesting topic. I found the approach of the wet and dry too simplistic, in fact despite the blinking can be seen as an average of 10 seconds dry, the authors never mention the fact that one side of the CL is always in contact with the tear film. at the end of the discussion you present some of the limitations (lipids protein, etc) but you don't mention this specific aspect. 

 It is now discussed at page 8, lines 303-312:

 Other factor that may also play role in vivo is that the posterior side of the CL is in permanent contact with the post-lens tear film [36].  Its capability to alter the hydration of the CL anterior side by diffusion of water through the hydrophobic silicon-rich lens core or by other mechanisms of tear exchange is hard to estimate as it greatly depends on the CL material properties and on the environmental conditions (air humidity, temperature, wind speed etc.) and is thought to be limited especially few hours after the fitting of the CL [36, 37]. It is found that the volume of the post-lens fluid squeezed out during the blink or lost due to pervaporation through the CL exceeds the amount of fluid drawn back under the lens at blink [36, 38]. This coincides with experimental findings, where the postlens TF thickness is greatly reduced in the course of SiHy wear [39, 40].

 2. I suggest to specify why you are using different contact angles measurements mode. Which information you get from each one of them? This part should be improved and references of the techiniques should be given.

 The Introduction now states at page 2, lines 66-75:

 The wettability of the samples is evaluated by the measurement of the advancing and receding water contact angles (CA) via (i) sessile drop and (ii) captive bubble (free and confined) techniques, both in static and dynamic (i.e. contraction and expansion of the bubble) modes. Advancing contact angle reports on the affinity of water to hydrate the surface of SiHy after pre-exposure to air (to desiccation), while the receding contact angle accesses the interaction (the “water holding” property of the CL) of the retreating water front with the hydrated CL surface [6, 12]. Water contact angles determined by various techniques have been shown to provide reliable in vitro estimate to the wettability of CLs with broad range of water content and polymer compositions [2, 4, 6, 12-17]. The capability of the different approaches to the measurement of CA (with drops or bubbles; in static or dynamic mode) to differentiate the studied SiHy materials is also analyzed.

 The Discussion section at page 7, lines 262- 273 now states:

 Various techniques are proposed to measure the advancing and receding water contact angles and were also adopted here: (i) from static measurements (sessile drop and free captive bubble) to (ii) dynamic measurements involving compression and expansion of the needle confined bubble [2, 4, 6, 12-17]. However there are almost no comparisons made on the capability of the different approaches to efficiently discriminate materials based on their distinct wettability. The advancing contact angle measures the affinity of an advancing water front to hydrate a surface pre-exposed to dryness (to air), i.e. it accesses the rehydration propensity of a material [6, 12]. The receding contact angle (measured when the water is forced to retract from a surface) evaluates the capability of a hydrated material to withhold water at exposure to dryness [6, 12]. It was found that the  advancing CA can be reliably evaluated via the sessile drop technique while in order to quantitatively estimate the receding CA dynamic measurements with needle confined captive bubble are need (the free bubble test provided qualitative albeit useful information).

 3. The difference in the wettability results are solely due to the chemistry of the material? what about the morphology, to make the work more complete, I suggest a morphological study of the surface ( SEM or AFM) to complete the results.

 In the recent years there is a high demand for simple and robust methods for in vitro and/or ex vivo quantitative estimation of CL wettability and the contact angle measurement methodologies become established as a reliable standard used as a standalone technique in multiple publications. Here are is a brief and non-exhaustive list of high profile references:

 Read ML, Morgan PB, Kelly JM, Maldonado-Codina C. Dynamic contact angle analysis of silicone hydrogel contact lenses. J Biomater Appl. 2011; 26(1):85-99. doi: 10.1177/0885328210363505.

 Campbell D, Carnell SM, Eden RJ Applicability of contact angle techniques used in the analysis of contact lenses. Eye Contact Lens. 2013; 39(3):254-62. doi: 10.1097/ICL.0b013e31828ca174.

 Svitova T.F., Lin M.C. Wettability conundrum: Discrepancies of soft contact lens performance in vitro and in vivo Eur. Phys. J. Spec. Top. (2011) 197: 295. doi:10.1140/epjst/e2011-01471-6.

 Maldonado-Codina C, Morgan PB. In vitro water wettability of silicone hydrogel contact lenses determined using the sessile drop and captive bubble techniques. J Biomed Mater Res A. 2007; 83(2):496-502.

 Lin MC, Svitova TF. Contact lenses wettability in vitro: effect of surface-active ingredients. Optom Vis Sci. 2010; 87(6):440-7. doi: 10.1097/OPX.0b013e3181dc9a1a.

 As stated in our reply to the previous point one of the goals of our study is to implement the variety of the contact angles measurement modes and to see what measurement mode and contact angle type will provide best discrimination between the CL materials. Indeed the high discriminative power of the advancing contact angle measurements via the sessile drop technique allows for a remarkably simple tool compared to the more experimentally demanding captive bubble setups. Thus the implementation of SEM or AFM methodologies is beyond the scope of the current study. Furthermore the experimental protocols of both methodologies may modify the surface of the contact lens compared to its state in the contact angle measurements.

However we reflected the potential impact of surface roughness in Discussion page 7, lines 251-261:

Apart from the chemistry of the SiHy, the structure of the CL surface might also impact its CA. Indeed although immediately after removal from the blister solutions CL materials are found to have a relatively smooth surface, the accumulation of the dehydration stress in the course of dehydration/rehydration cycling can also modify the roughness of the CLs, which has strong impact on wettability by itself [32, 33]. The effect can be additionally enhanced in vivo where the deposits of tear film constituents (lipids, proteins, mucins etc.) are well known to alter both the composition and the structure of the CL surface [34], and correlates with our findings on very different TF breakup patterns of SiHy with internal wetting agents and no surface coating and on contact lens with true surface coating (representative patterns are provided at point III in Supplement 1). This suggests that various effects, including both surface chemistry and alterations in surface roughness, may simultaneously contribute to the wettability of the CL surface.

 4. Line 257 , specify the reference ("previous studies")

References [12, 22] are included at line 325.

5. Line 258 did you verify that the 24 hours were enough for the "washing"? 

 The surface tension of the washing solution was monitored. If there are wetting agents desorbing from the CL to the solution this will decrease its surface tension as described previously. We have found that after 1 h the surface tension of the washing solution was the same as of pure water (~72.9 mN/m). The CL perfusion was extended to 24 hour to ensure the removal of trace amounts that might be present in the CL material. Control tests have shown no difference in the contact angles of any of the SiHy materials after 24 h and 48 hours of washing. This is now mentioned at page 9, lines 328-330 of the revised manuscript. 

 6. I suggest to move table 1 and 2 to anexe.

 This is now done. Instead in order to statistically compare the differences in the advancing and receding contact angles observed between the different pairs of CLs specimen at the different time points, Cohen’s d was utilized as recommended for the analysis of laboratory or clinical properties of biomaterials (Ratner, B.D., Correlation, Materials Properties, Statistics and Biomaterials Science, Chapter III.1.3 in Biomaterials Science (Third Edition) An Introduction to Materials in Medicine, Editors: Buddy D. Ratner, Allan S. Hoffman, Frederick J. Schoen, Jack E. Lemons, 2013, pp: 1354 and others). Cohen’s d represents a measure of the effect size, i.e. the normalized difference between the mean wettability of two samples, which accounts not only for the differences in the advancing CA, but also for the sample size and the magnitude of the noise (the random error) among the individual measurements within a sample. Thus it allows for visual and numerical estimation between the differences in the properties of each pair of SiHy materials and also for evaluation of the significance of the observed effect size (d≥0.8- large; d= 0.5- medium; d= 0.2- small).

The analysis is presented at lines 99-129 and lines 173-181 and as Fig. 2 and Fig. 4 in the revised manuscript.

 7. figure 6 is misplaced. 

 The referencing of the figure in the text and its number are now synchronized.

 8. I suggest to add more references of previous studies dealing with silicon CL wettability in the introduction and in the results discussion. There are several of them (example " Controlled Release of Antibiotics From Vitamin E Loaded Silicone-Hydrogel Contact Lenses " they measure the wettability of AOD and TE, and their results are close to yours for example). I suggest the insertion of More references on what concerns the adhesion energy.

 The publication is now included as reference [29]. Number of other references were implemented including some recent reviews, i.e. References [6, 12, 14, 17, 21, ect], which offer perspective of the published research to the interested reader. References 23-25 provide detailed information on adhesion energy theory.

Round  2

Reviewer 3 Report

Thank you for your answers, I believe that now your work is improved and ready for publication